# Ribosomes: The New Role of Ribosomal Proteins as Natural Antimicrobials

**DOI:** 10.3390/ijms23169123

**Published:** 2022-08-14

**Authors:** Jessica J. Hurtado-Rios, Ulises Carrasco-Navarro, Julio Cesar Almanza-Pérez, Edith Ponce-Alquicira

**Affiliations:** 1Departamento de Biotecnología, Universidad Autónoma Metropolitana Unidad Iztapalapa, Av. San Rafael Atlixco 186, Col. Vicentina, Ciudad de México 09340, Mexico; 2Departamento de Ciencias de la Salud, Universidad Autónoma Metropolitana Unidad Iztapalapa, Av. San Rafael Atlixco 186, Col. Vicentina, Ciudad de México 09340, Mexico

**Keywords:** ribosome, ribosomal protein, antimicrobial peptides, moonlighting protein

## Abstract

Moonlighting proteins are those capable of performing more than one biochemical or biophysical function within the same polypeptide chain. They have been a recent focus of research due to their potential applications in the health, pharmacological, and nutritional sciences. Among them, some ribosomal proteins involved in assembly and protein translation have also shown other functionalities, including inhibiting infectious bacteria, viruses, parasites, fungi, and tumor cells. Therefore, they may be considered antimicrobial peptides (AMPs). However, information regarding the mechanism of action of ribosomal proteins as AMPs is not yet fully understood. Researchers have suggested that the antimicrobial activity of ribosomal proteins may be associated with an increase in intracellular reactive oxidative species (ROS) in target cells, which, in turn, could affect membrane integrity and cause their inactivation and death. Moreover, the global overuse of antibiotics has resulted in an increase in pathogenic bacteria resistant to common antibiotics. Therefore, AMPs such as ribosomal proteins may have potential applications in the pharmaceutical and food industries in the place of antibiotics. This article provides an overview of the potential roles of ribosomes and AMP ribosomal proteins in conjunction with their potential applications.

## 1. Introduction

The ribosome is an organelle within the cytoplasm, implicated in protein translation, and present in the three domains of life: bacteria, eukarya, and archaea. Ribosomes are composed of two subunits, one large and one small. Each of them is composed of proteins and ribosomal ribonucleic acid (rRNA). Recently, some ribosomal proteins have shown antimicrobial activity; therefore, they could be a new source of antimicrobial peptides (AMPs). Recently, many proteins have been identified as “moonlighting”, when the same protein plays two or more unrelated functions in a cell or organism [1,2]. Interest from the scientific community has increased as these AMPs have shown a broad-spectrum antimicrobial response against infectious bacteria (Gram-positive and Gram-negative), viruses, parasites, fungi, and tumor cells [3].

Therefore, the objective of this review is to present an overview of the composition, function, and structure of ribosomes, as well as an update regarding current research into their antimicrobial activity, whether obtained from natural sources (e.g., secreted by microorganisms) or obtained synthetically. These ribosomal proteins, as natural antibiotics, could provide a solution to increasing antibiotic resistance worldwide. Finally, we encourage further investigation into the possible mechanisms of action of the antimicrobial activity of AMPs.

## 2. Ribosome

Particles containing ribonucleic acid (RNA), which were probably ribosomes, were first observed and reported during the 1940s. Within the following decades, several studies yielded essential data, such as the role of ribosomes for protein synthesis and the differentiation of eukaryotic and prokaryotic ribosomes, the latter being smaller [4].

### 2.1. Composition

Ribosomes are macromolecular complexes located in the cytoplasm of living cells and are present in the three domains of life: bacteria, eukarya, and archaea. Their main role in the cells is the synthesis of proteins by translation, which consists of decoding messenger ribonucleic acids RNA (mRNAs) into proteins. The composition of ribosomes is somewhat similar among the domains; they are composed of two subunits, a large (L) subunit and a small (S) subunit, both of which contain rRNA (Table 1), and ribosomal proteins. The most common way to distinguish ribosomes from different domains of life has been through the measure (S) or the Svedberg unit (Sv), in addition to the relative sizes of their subunits [1,4,5,6].

In recent years, a database of ribosomal protein sequences has been created and expanded due to the number of genomes that have been sequenced, identifying approximately 53 ribosomal proteins in bacteria, 58 in archaea, and 78 in eukarya. Based on the analysis and comparison of these genomes, more than 50% of bacterial ribosomal proteins have homologous proteins in eukarya and archaea (as shown in Figure 1). Table 2 shows each of the ribosomal proteins identified in the three domains of life. This comparison was conducted in reference to *E. coli*, in which the small ribosomal subunit contains proteins S1–S21, and the large subunit contains proteins L1–L36. Specific proteins were deleted, such as: L7, a modified form of L12 found only in a limited species; L8, a complex of L7/L12 and L10; and L26, identical to S20. Lastly, every ribosome has only one copy of the proteins, except for protein L7/L12 in *E. coli*, in which there are four copies per ribosome [9,10].

### 2.2. Nomenclature

Initially, the nomenclature of ribosomal proteins was designed based on the organism, in accordance with their electrophoretic separation, and designed with an S or L, indicating whether they are part of the small or large ribosomal subunit, respectively [12]. Nevertheless, this nomenclature could be misinterpreted because the same numbers corresponded to homologous ribosomal proteins of different species. Therefore, comparison of the amino acid sequences of these proteins was key to discerning between the homology of ribosomal proteins of different species and demonstrating the evolutionary conservation of most of these proteins. The ribosomal proteins from *E. coli* were the first to be isolated and fully sequenced; therefore, their archaeal and eukaryotic homologs were assigned *E. coli* names. Based on this, some authors have established a new nomenclature [13] (Table 2), which is based on the assignment of homologous ribosomal proteins with the same name, irrespective of species, followed by the prefixes “b”, “a”, and “e”. The prefix “b” (from “bacterial”) is assigned to bacterial ribosomal proteins without eukaryotic or archaeal homologs. Similarly, archaeal ribosomal proteins having no homologs in bacterial and eukaryotic ribosomes are designated with the prefix “a” (from “archaeal”). Lastly, the prefix “e” (from “eukaryotic”) is assigned not only to eukaryotic ribosomal proteins without bacterial or archaeal homologs, but also to the homologous archaeal proteins. However, incorporating the prefix “u” indicates that the protein is present in the ribosomes of the three domains of life [14]. Moreover, it is also possible to find cited ribosomal proteins in the literature, such as LSU or SSU, referred to as large and small subunits, respectively; some authors have considered this, such as Londei and Ferreira-Cerca [10].

**Table 2 ijms-23-09123-t002:** Ribosomal proteins classified by taxonomic range based on sequence and structural information [4,9,13,15].

**Ribosomes Proteins of the Small Subunit**								
**Family Name**	**Alt. Name ^1^**	**Taxonomic Range**				**Univ. Cons. ^2^**	**Deleted Mutant**	**Ribosomal Function/Comment**
		**A**	**B**	**E**	**O**			
bS1		○	●	○	C		X	Brings with mRNA into the proximity of the ribosome during initiation; involvement in translational regulation [16,17,18]
uS2	S0Ae	●	●	●	C, M	1		Involvement in translational regulation [17]
uS3	S3e	●	●	●	C, M	1		Forms the mRNA entry pore and may exhibit helicase activity to unwind mRNA secondary structures encountered during translation [18]
uS4	S9e	●	●	●	C			Mutations (ram) increase the error during the decoding process; with uS7 as one of the two assembly initiator proteins; involvement in translational regulation [16,17,18]
uS5	S2e	●	●	●	C, M, T	1		Probably facilitates changes in rRNA conformation that alter the selection mode of the ribosome from accurate to error-prone and vice versa [16,17,18]
bS6		○	●	○	C, M, T		X	Form a tight complex that assembles as a heterodimer on the outer edge of the platform of the subunits [16]
uS7	S5e	●	●	●	C, M	1		mRNA and tRNA binding at the E site; involvement in translational regulation [16,17,19]
uS8	S22e	●	●	●	C, T	1		Involvement in translational regulation [17]
uS9	S16e	●	●	●	C, M, T	1	X	Interaction with P site [19]
uS10	S20e	●	●	●	C, M	1		uS3, uS10, and uS14 form a tight protein cluster at the back of the 30S head [16]
uS11	S14e	●	●	●	C, M, T	1	X	Forms part of the binding site of the anticodon loop of E-tRNA [19]
uS12	S23e	●	●	●	C, M	1		Involved in decoding of the coding of the second and third codon positions at the A site [16,17,18,19]
uS13	S18e	●	●	●	C		X	Interaction with P site tRNA [19]
uS14	S29e	●	●	●	C, M	1		uS3, uS10, and uS14 form a tight protein cluster at the back of the 30S head [16]
uS15	S13e	●	●	●	C, M, T	1		Involvement in translational regulation [17]
bS16		○	●	○	C, M, T			Improve the stability of ribosome [16]
uS17	S11e	●	●	●	C, M, T	1	X	Ribosome assembly [20]
bS18		○	●	○	C, M, T			Form a tight complex that assembles as heterodimer on the outer edge of the platform [16]
uS19	S15e	●	●	●	C			Inter-subunit bridges; related to the changes in the information between LSU and SSU [20]
bS20		○	●	○	C		X	Involvement in translational regulation [17]
bS21		○	●	○	M			Required for the recognition of native templates; stabilize the ribosome [21]
bS22		○	●	○	○			Accumulates in ribosomes of the stationary phase, so is a factor rather than an RP [17]
bTHX		○	●	○	○			Found in thermophilic bacteria and stabilizes the organization of RNA elements at the 30S subunit [21]
eS1		●	○	●	○			Initiation of translation by catching the mRNA and directing it to the ribosome; involvement in translational regulation [17,18]
eS4		●	○	●	○			Assembly initiator protein; involvement in translational regulation [16,17]
eS6		●	○	●	○			Form a tight complex that assembles as heterodimer on the outer edge of the platform [16]
eS7		○	○	●	○			mRNA and tRNA binding at the E site; involvement in translational regulation [16,19]
eS8		●	○	●	○			Involvement in translational regulation [17]
eS10		○	○	●	○			Associated ribosome quality control; stall translation on poly(A) sequences [22]
eS12		○	○	●	○			tRNA decoding at the A site [19]
eS17		●	○	●	○			Ribosome assembly [20]
eS19		●	○	●	○			Inter-subunit bridges [20]
eS21		○	○	●	○			Required for the recognition of native templates and its function resembles the function of bS1 [21]
eS24		●	○	●	○			Required for processing of pre-RNA and maturation of SSU [23]
eS25		●	○	●	○			Translation of mRNAs from specific cellular pathways [24]
eS26		○	○	●	○			Involved in the formation of the mRNA binding channel in the region of the exit site [25]
eS27		●	○	●	○			Fused to C-terminus of ubiquitin; zinc finger [4]
eS28		●	○	●	○			mRNA interactions [4]
eS30		●	○	●	○			Replaces part of bS4 [4]
eS31		●	○	●	○			Ribosome assembly [26]
RACK1		○	○	●	○			Interacts with signaling molecules; through this interaction, the regulation of translation is mediated [18,19]
**Ribosomal Proteins of the Large Subunit**								
**Family Name**	**Alt. Name ^1^**	**Taxonomic Range**				**Univ. Cons. ^2^**	**Deleted Mutant**	**Ribosomal Function/Comment**
		**A**	**B**	**E**	**O**			
uL1	L10a	●	●	●	C, M	1	X	Possibly involved in the disposal of deacylated tRNA that has been release from the E site; translational feedback regulation of L11 operon; involvement in translational regulation [16,17,18]
uL2	L2e	●	●	●	C, M, T	1		Residue histidine 229 is possibly involved in the peptidyltransferase reaction [17,19]
uL3	L3e	●	●	●	C, M, T	1		Assembly initiation: forms an assembly starting point at the 3’region of 23S rRNA of the LSU; plays a role in the allosteric coordination of the peptidyl transferase center (PTC) [16,17,19]
uL4	L4e	●	●	●	C, M, T	1		Role in rRNA transcription antitermination; involvement in translational regulation [16,17,18]
uL5	L11e	●	●	●	C			With uS13, forms different contacts in the rotational states of ribosome; interaction with P site tRNA [19]
uL6	L9e	●	●	●	C			Forms the factor-binding site at the edge of the inter-subunit cleft of the ribosome [19]
bL9		○	●	○	C, M, T			Mutations might affect the precise arrangement of a tRNA in the P site [17,18]
uL10	P0	●	●	●	M			Involvement in translational regulation [17,19]
uL11	L12e	●	●	●	C, M, T	1	X	During the stringent response, L11 senses the presence of a deacylated tRNA in the A site [16,17,18]
bL12		○	●	○	C, M			Involved in elongation-factor binding, possibly in GTPase activation and in translational regulation [16,17,18]
uL13	L16e	●	●	●	C, M, T	1		Can repress rRNA transcription termination [27]
uL14	L23e	●	●	●	C, M, T	1		Forms the factor-binding site at the edge of the inter-subunit cleft of the ribosome [19]
uL15	L28e	●	●	●	M, T		X	Essential late assembly step that is important for an active ribosomal conformation; improve the stability of ribosomes [16,17]
uL16	L10e	●	●	●	C, M, T	1		Essential late assembly step. May be involved in correct positioning of the acceptor stem of A and P site tRNA as well as ribosome recycling factor (RRF) on the ribosome; acts in information transmission of the eukaryotic ribosome [16,17,18,26]
bL17		○	●	○	M, T			Forms a ring around the tunnel exit site [18]
uL18	L5e	●	●	●	C, M	1		With L5 and L25, forms a complex with 5S rRNA which constitutes the central protuberance of the SSU [17]
bL19		○	●	○	C, M		X	Inter-subunit bridge in the ribosome [20]
bL20		○	●	○	C, M, T			Involvement in translational regulation; improve the stability of ribosomes [17]
bL21		○	●	○	C, M, T			Direct contact with the 23S rRNA [26,28]
uL22	L17e	●	●	●	C, M, T	1		May interact with specific nascent chains to regulate translation [16,17,18]
uL23	L25e	●	●	●	C, M, T	1		Present at the tunnel exit site and has been shown to be a component of the chaperone trigger factor binding site on the ribosome [16,18]
uL24	L26e	●	●	●	C, M, T	1	X	Important for assembly initiation; improve the stability of ribosomes [16,17]
bL25		○	●	○	○			With L5 and L18 forms a complex with 5S rRNA, that constitutes the central protuberance of the SSU [17]
bL27		○	●	○	C, M, T		X	Implicated in the placement of the acceptor stem of P site binding of the ribosome recycling factor on the 50S subunit [16,18]
bL28		○	●	○	C, M, T		X	Assembly protein [20]
uL29	L35e	●	●	●	C, M, T	1	X	Located close to the tunnel exit site and may constitute part of the binding site for the signal recognition particle [16,18,26]
uL30	L7e	●	●	●	M, T		X	Assembly of the bacterial SSU or the eukaryotic LSU [26]
bL31		○	●	○	C			Contributes to ribosome subunit association [20,28,29]
bL32		○	●	○	C, M			Forms part of the tunnel near to the peptidyl transferase center [18]
bL33		○	●	○	C, M, T		X	Assembly protein [20]
bL34		○	●	○	○			Ribosome-constituting protein [30]
bL35		○	●	○	C, M			Assembly protein [20]
bL36		○	●	○	C, M			Assembly protein [20]
P1/P2		●	○	●	○			Mediate elongation factor GTPase activity [16]
eL6		○	○	●	○			Part of the peptidyl transferase center [20]
eL8	L7a	●	○	●	○			Assembly of the bacterial SSU or the eukaryotic LSU [26]
eL13		●	○	●	○			Can repress rRNA transcription termination; forms bridges between LSU and SSU [15]
eL14		●	○	●	○			Role in GAC; forms part from the elongation factors to the mRNA exit tunnel [20]
eL15		●	○	●	○			Improves the stability of ribosomes [16,17]
eL18		●	○	●	○			Forms tetrameric complex with 5S rRNA [17]
eL19		●	○	●	○			Peptide exit tunnel; participates in inter-subunit bridges [20,26],
eL20		○	○	●	○			Improves the stability of ribosomes [16]
eL21		●	○	●	○			Bridges functional sites: peptidyl transferase center, the tunnel, and a tRNA binding site [26]
eL22		○	○	●	○			Ribosome assembly and protein translation [16,19]
eL24	L10a	●	○	●	○			Improves the stability of ribosomes [17]
eL27		○	○	●	○			Binding of tRNA to the ribosome [16]
eL28		○	○	●	○			Assembly protein [20]
eL29		○	○	●	○			Assembly of the SSU [26]
eL30		●	○	●	○			Assembly of the SSU [26]
eL31		●	○	●	○			Contributes to ribosome subunit association [29]
eL32		●	○	●	○			Forms part of the tunnel exit site [18]
eL33		●	○	●	○			Assembly protein; interacts directly with E site tRNA [20,28]
eL34		●	○	●	○			Ribosome-constituting protein [30]
eL36		○	○	●	○			Assembly protein [20]
eL37		●	○	●	○			Structural constituent of ribosome; binds to the 23S rRNA [28]
eL38		●	○	●	○			Subunit of the cytosolic LSU; involved in translation [31]
eL39		●	○	●	○			Lines the tunnel and gives it its own “Teflon-like” properties [19]
eL40		●	○	●	○			Enables protein binding and ubiquitin protein ligase binding [32]
eL41		●	○	●	○			Interacts with beta subunit of protein kinase CKII and stimulates phosphorylation of DNA topoisomerase [33]
eL42	L44a	●	○	●	○			Enables RNA binding and structural constituent of ribosome [34]
eL43	L37Ae	●	○	●	○			Enables RNA binding and structural constituent of ribosome [35]

^1^ Alternative name. Bairoch; http://www.expasy.org/search/ribosomal%20proteins (accessed on 16 May 2022). ^2^ Universal conservation. Proteins found in all groups of genomes, including mitochondria and chloroplasts. L, large subunit; S, small subunit; A, archaea; B, bacteria; O, taxonomic range; C, chloroplast; M, human mitochondria; T, trypanosome mitochondria; L7b = L12b; L8b = L10b: (L12b), n = 4 or 6; L16b = L10e; L26b does not exist = S20b.

### 2.3. Structure

Given the main cellular functions of ribosomes, it is essential to understand their structure; nonetheless, their structure has been difficult to fully describe, due to their size and lack of symmetry. Many methods have been developed to probe the ribosomal structure, particularly regarding the binding sites for mRNA and transfer RNA (tRNA). The methods commonly used to analyze ribosomal structures have included electron microscopy (EM). In the 1970s, this method provided low-resolution images from negatively stained samples of the shapes and structures of the subunits of ribosomes. Nevertheless, even though the resolutions of EM have improved, X-ray crystallography has provided atomic details of ribosomes. Currently, the methods that have provided more information about ribosomal structures are cryo-electron microscopy and macromolecular crystallography. These methods have clarified ribosomal structures, providing fundamental knowledge regarding the structural basis of their translation mechanisms [4,36].

The overall composition and structure of ribosomes are based on RNA molecules, which shape ribosomal subunits. Additionally, multisystem RNA junctions and tight bends in the RNA interact with proteins, which, in turn, are distributed on the back and sides of ribosomal subunits, rather than being buried in the middle of subunits; therefore, the ribosome literally has an RNA nucleus. There are differences in the structures of the 30S and 50S subunit ribosomes of prokaryotes, which are based on the domains of the secondary structure of 23S rRNA interacting to form a single, almost hemispherical mass. In contrast, the domains of the secondary structure of 16S rRNA are independent, and therefore have few interactions with each other [7,36].

### 2.4. Role of Ribosomes in the Cell Translation Process

Deoxyribonucleic acid (DNA) is usually found as linear chromosomes in eukaryotes and circular chromosomes in prokaryotes, and the set of chromosomes in a cell comprise its genome. The genes carried by DNA encode the protein molecules. The gene expression, when it manufactures its corresponding protein, has two steps. The first consists of the transcription process, in which the information in DNA is transferred to an mRNA molecule. The last step in gene expression is translation [37] (Figure 2), and which is described in detail in the following paragraphs.

Translation is the process by which a protein is synthesized from the information encoded in a molecule of mRNA into the polypeptide chains of amino acids which comprise blocks of proteins; the genetic code, i.e., trinucleotide combinations, referred to as three-letter codons, corresponds to a specific amino acid or stop signal. This process occurs in the peptidyl transferase center (PTC), where the ribosome promotes the joint binding and reading of mRNA and tRNA via interactions between its small and large ribosomal subunits, as well as other translation factors [38]. In eukaryotic organisms, mature mRNA molecules exit the nucleus and travel to the cytoplasm, where the ribosomes are located. Translation occurs in three stages: initiation, elongation, and termination. At the initiation stage, the translation begins at the 5′end of the mRNA, while the 3′ is still attached to the DNA. At this stage, the ribosome containing three binding sites, an amino acid site (A), a polypeptide site (P), and an exit site (E), meets, and the first tRNA molecule carrying the amino acid methionine binds to the region near the start codon in the mRNA (Figure 3) [37,39].

The most common starting codon for known *E. coli* genes is AUG in 83% of genes; however, over the years, many studies have demonstrated alternative start codons, such as GUG and UUG, in 14% and 3% of genes, respectively. The near region where ribosomes bind in mRNA corresponds to particular amino acids, and is better known as the untranslated region (UTR), or leader sequence. In bacteria, 5´UTR is short and known as the Shine–Dalgarno box (AGGGAGG). A similar region in eukaryotic mRNA was characterized as the Kozak box (ACCAUGG) [39,40].

The next stage in translation is elongation, which involves repetitive decoding cycles, peptide bond formations, and translocation, so that the tRNAs bring the amino acids to the ribosome, joining together to form the amino acid sequence (polypeptide chain) of a protein [37,41].

Lastly, in bacteria, the termination stage entails two steps. The first step corresponds to the recognition of the stop codon. This occurs when the termination factors RF1 and RF7 recognize a stop codon UAG, and UAA or UGA and UAA, respectively. The second step occurs with the hydrolysis of the ester bond of the peptidyl-tRNA. Once the ribosome recognizes that the translation is complete, the new protein is created [41,42]. The main difference in bacteria is that the transcription and translation occur simultaneously, and the mRNAs are short-lived. Compared with eukaryotes, mRNAs have half-lives and are subject to modification, and they must exit the nucleus to be translated [37,41].

## 3. Moonlighting Proteins

Literature reviews by Chen et al. (2021) [2] and Sing and Bhalla (2020) [43] mentioned the first moonlighting proteins (MLPs). One described MLPs as proteins that have multiple tasks in a cell or organism, whereas the other referred to MLPs as the same protein expressed from the same gene performing two or more unrelated functions in a cell or organism. Currently, MLPs are considered fairly common, because a significant number of proteins have been reported and more than 350 MLPs have been identified. For practical purposes, not all MLPs are described in this article. Therefore, for more information, Table 3 presents the databases that track the increasing number of MLPs discovered [44].

Several assumptions have been postulated to explain how MLPs might evolve. First, a protein can be an MLP if it exhibits an interaction with another molecule that produces a benefit. Second, it has been hypothesized that gene duplication promotes the development of the MLP function. Third, MLPs increase mutational robustness by enabling biological systems to traverse new landscapes not predicted in natural selection [43,44].

The relationship between MLPs and other molecules is a well-defined hypothesis in the literature that has become an accepted assumption, likely due to the mutually beneficial nature of their interactions for the organism. MLPs conserve energy, which is a limited resource in any system, by delegating two functions to a single protein. In addition, the two functions are typically related [44].

The general structure and physical properties of proteins are appropriate to allow them to carry out their primary function and bind to a substrate and catalyze a reaction. During the evolution of proteins over the years, no significant changes in their structure were required to allow the protein to adapt to perform different functions [44]. However, it has been found that some MLPs change their function by undergoing large conformational changes or transitions between intrinsically unfolded domains and multiple folded structures, so that different conformations of the protein structure can perform different functions. In the case of some intracellular/extracellular MLPs, large changes in their structure or conformation are not necessary to perform the extracellular function. In many, the extracellular function involves binding to another molecule, often another protein. If catalysis occurs when the protein is inside the cell, the three-dimensional structure of an enzyme or chaperone includes a large amount of solvent-exposed surface area, and some portion of this surface can gain a pattern of amino acids needed for interacting with another molecule, a cell surface receptor, or another protein, without affecting the active site where catalysis occurs [46].

Once the correct level of protein is reached, each of the functions of an MLP needs to be performed at the appropriate level. Some MLPs can perform multiple functions at the same time; in other cases, their different functions are regulated so that the MLP performs only one function at a time and can change to a different function when conditions within the cell change. In 2017, Jeffery C.J. [47] suggested that post-translational modifications (PTMs) are normally used to regulate protein function in general, and that they might be used to induce an MLP to switch to a different function. PTMs involve many types of alterations to the polypeptide chain, including the removal or addition of a functional group such as a phosphoryl, acyl, or acetyl group. For example, the addition of a phosphate group can result in considerable changes in the conformation of a protein and its ability to bind to other molecules. In some cases, PTMs can serve as a signal “zip code” (postal code) to target the protein to another location in the cell. Some examples of the above are when several proteins of the ribosome that have been phosphorylated leave the ribosome to participate in other activities. It has been shown that ribosomal protein (RP) uS3 in bacteria, when phosphorylated, leaves the ribosome and participates in DNA damage repair, as well as acting as a transcription factor. In the same way, when the RP eL13 is phosphorylated, this RP moves to join a multiprotein transcription factor in the nucleus. Finally, some authors have proposed that mutations in gene-encoding transcription factors can also alter the regulation of MLP expression [46,47,48].

In the case of intracellular MLPs, these proteins also play significant intracellular functions; therefore, the secretion of these proteins is closely regulated and is usually induced by specific stimuli. MLPs need a signal peptide to be exported; in contrast, without a signal peptide, these MLPs are not immediately secreted after translation and are not efficiently secreted. Furthermore, these proteins are normally secreted during the stationary phase. However, regarding intracellular/surface MLPs that share physical–chemical characteristics with other cytoplasmatic proteins, their selected secretion, how they cross the cell membrane, and how they become attached to the cell surface are unknown [47,49].

MLPs have been documented from prokaryotes to eukaryotes, and their primary or original function is the canonical function. Figure 4 shows the categories of MLPs based on the secondary and canonical functions. One of the first MLPs discovered was lactate dehydrogenase, which adopted an additional function as a structural protein in the eye lens (crystallins). Other examples include enolase from *Streptococcus pneumoniae*, which is a virulence factor in addition to being a glycolytic enzyme, and malate synthase (MS), which has a significant role in the glyoxylate pathway and is also a virulence factor. The glyceraldehyde-3-phosphate dehydrogenase (GAPDH) of *E. coli* has different functions at the same catalytic site, including as a housekeeping protein and promoting NAD+-dependent ADP ribosylation, which is involved in host–pathogen interactions. Furthermore, this molecule supports peptidoglycan as a ligand to surface GAPDH [43].

### Ribosomal Proteins as Moonlighting Proteins

Hundreds of MLPs have been discovered in several prokaryotes and eukaryotes; prokaryotes are involved in virulence, whereas eukaryotes are involved in pathogen virulence and attachment. Bacterial MLPs have also been observed as homologs to human proteins, such as chaperonins, aldolases, GAPDH, and enolases [43,44]. In 1996, Wool [50] proposed the hypothesis that many RPs have a second function in addition to their main role. Over the last decades, dozens of ribosomal proteins (RPs) whose canonical functions are as transcription or translation factors have been identified as MLPs [2,43]. The moonlighting functions of these MLPs in eukaryotic cells include DNA repair, the autogenous regulation of translation, inhibiting MDM2-mediated p3 ubiquitination and degradation, inhibiting the splicing of its own RNA transcript, inhibiting the removal of intron, binding to and inhibiting HDM2 protein (a key negative regulator of the tumor suppressor p53), regulating (increasing) the translation of p53 tumor suppressor protein after DNA damage, supporting abiotic stress resistance, shortening the half-life of its own mRNA, binding DNA stimulating the unwinding of DNA by the Rep helicase protein and being involved in the NSP-interacting kinase (NIK) receptor-mediated defense pathway to defend against the Gemini virus substrate, and being the binding partner of NIK1, to mention some examples (for a more detailed review, check the MLP databases in Table 3 [43,44,45]. Remarkably, these RPs have adapted an antimicrobial function as AMPs [1,2].

AMPs consist of molecules that inhibit numerous organisms and multiple pathogens, such as bacteria, fungi, viruses, and parasites. Recent research has identified the importance of AMPs as an alternative to traditional antibiotics, due to the developed bacterial resistance against various classes of traditional antibiotics. In addition, AMPs are more stable and tend to develop less resistance in the innate immune system; thus, their application could be essential for use in medicine, food, etc. However, their complicated extraction process and low production yield have limited their potential applications and scientific research [51,52].

As previously mentioned, in recent decades, various RPs have been identified as AMPs. Several articles have referred to the first reported RP with antimicrobial activity in 1999, where Pütsep et al. [53] synthesized one peptide of RP uL1 from *Helicobacter pylori* that had indicated antimicrobial activity. In 2002, two other RPs were reported in the literature as AMPs. The first, described by Fernandes and Smith [54], was purified from skin secretions and trout epithelial cells. The antimicrobial peptide was identified as 40S ribosomal protein S30 (RP eS30) through matrix-assisted laser desorption/ionization time-of-flight (MALDI-TOF) mass spectrometry, with a single signal of 6.6 kDa. The RP eS30 presented antimicrobial activity against *Planococcus citreus*, which is considered to be a Gram-positive pathogen commonly found in intertidal sediments. The second RP reported in 2002 by Lee et al. [55] was a fragment of the RP uL1 (peptide HP 2–20), derived from the N-terminus of *H. pylori* ribosomal protein L1, which exerts antimicrobial activity against *Candida albicans*, a pathogen that causes disease in humans.

Another remarkable fact regarding RPs as AMPs is that they have been produced by microorganisms present in certain foods. These microorganisms are classified as lactic acid bacteria (LAB), which are defined as a class of Gram-positive bacteria that do not form spores and whose main product of fermented sugar is lactic acid. LAB are considered to be a type of probiotic due to their health-promoting effects on the host, and are effective in treating human and animal diseases [56]. The first case of 30S ribosomal protein S21 (RP bS21) from *Lactobacillus sakei* was isolated from a Brazilian meat product. The RP bS21, identified by de Carvalho et al. [57], showed antimicrobial activity against *Listeria monocytogenes*. This microorganism is considered to be an important food pathogen that causes severe infection listeriosis. Additional RPs were identified by Pidutti et al. [58], the L27 and L30 proteins of the 50S ribosomal, RP bL27 and RP uL30, respectively. The LAB that produces these RPs were identified as *Lactobacillus salivarius*, isolated from the feces of four-month-old human infants. These RPs exerted antimicrobial activity against *Streptococcus pyogenes*, *Streptococcus uberis*, and *Enterococcus faecium*. The first two microorganisms are bacterial pathogens that cause mild to life-threatening infections, whereas the third microorganism, despite being commensal in the human gut, is a pathogen that causes diseases such as neonatal meningitis [59].

In 2019, García-Cano et al. [60] reported more LAB that produce the 50S ribosomal protein L36 (RP bL36) of *Pediococcus acidilactici*, isolated from Gouda cheese. RP bL36 displayed antimicrobial activity against *E. coli* and *Listeria innocua*, which are considered to be pathogenic food microorganisms. To date, the last RP reported as an AMP was described by Ghoreishi et al. [61]. They detected the 50S ribosomal protein L1 (RP uL1), which was produced by *Bacillus tequilensis* isolated from healthy human feces. This RP showed an antibacterial effect against *Staphylococcus aureus*, an important food-poisoning pathogen. Table 4 lists the RPs identified in the literature, to date; the RPs that have been identified from their production by microorganisms are listed regardless of the genus or species. Moreover, the physicochemical properties are also listed, obtained from the pepcalc.com peptide property calculator (https://pepcalc.com/) (accessed on 21 June 2022)and the Biosynthesis peptide property calculator (https://www.biosyn.com/peptidepropertycalculator/peptidepropertycalculator.aspx) (accessed on 21 June 2022) databases.

The gene ontology (GO) analysis of each RP in Table 4 and Table 5 was realized using the UniProt Knowledgebase website (https://www.uniprot.org/, accessed on 21 June 2022). GO provides a structured, organized biological database of the genes potentially present in all organisms, and provides a specific definition of the protein functions [62]. GO considers three distinct aspects of gene functions and is, therefore, subdivided into three non-overlapping ontologies: molecular function (MF), cellular component (CC), and biological process (BP). The MF is a molecular-level process or activity that can be performed by the action of a single macromolecular machine. The CC is a specific location relative to the cell compartments and structures that are occupied by a macromolecular machine when it conducts an MF (CC). Lastly, the BP represents the specific objective for which the organism is genetically programmed; each biological process is described by its results or end state. For example, in the process of cell division, the result is the creation of two daughter cells (i.e., a divided cell) from a single parent cell [63,64]. The results obtained from the GO analysis for all the listed RPs were as follows: the MF was as a structural constituent of the ribosome and tRNA binding; the CC was not found; and the BP was translation and the regulation of translation.

**Table 4 ijms-23-09123-t004:** Different ribosomal proteins as AMPs isolated from organisms.

Ribosomal Protein as AMP	Peptide Sequence	Isolated From	Against Microorganism	Physiochemical Properties		
				Hydrophobic	Iso-Electric Point (pH)	Net Charge at pH 7
RP eS30 6.6 kDa [54]	KVHGSLARAGK	*Oncorhynchus mykiss*	*P. citreus* and *Bacillus subtilis*	36.36%	11.57	3.1
RP bS21 6.7 kDa [57]	GKTVVRSNESLDDALRRFKRSVSKAGTIQEYRKR	*L. sakei*	*Enterococcus faecalis*, *L. sakei*, *L. innocua*, *L. monocytogenes*, *Listeria seeligeri*, and *Staphylococcus epidermidis*	29.41%	11.31	6
RP uL1 24.6 kDa [65]	----	*Lactobacillus* Hma2N	*Melissococcus plutonius*	----	----	----
RP eL29 6.4 kDa [66]	AKSKNHTSHNQNRKQHRNGIHRPKTYRYPSMKGVDPKFLKNLKFSKKHNKNTKK	*Crassostrea gigas*	*B. subtilis*, *E. coli* and *Vibrio parahaemolyticus*	18.52%	11.71	16.5
RP bL27 9.9 kDa [58]	----	*L. salivarius*	*S. pyogenes*, *S. uberis*, and *E. faecium*	----	----	----
RP bL36 4.4 kDa [60]	MKVRPSVKPMCEHCKIIKRKGRVMVICSANPKHKQRQGK	*P. acidilactici* OSU-PECh-3A	*E. coli*	35.9%	11.3	11
RP eL27 4.1 kDa [67]	PALKRKARREAKVKFEXRYXTGXNXXFFQ	*Silurus asotus*	*B. subtilis*, *S. aureus*, *Micrococcus luteus*, and *Streptococcus iniae*	37.5%	11.3	11
RP uL1 15 kDa [61]	----	*B. tequilensis*	*S. aureus*	----	----	----

Due to the interest in new antimicrobial substances for multiple applications in the medical and food industries, researchers have been working on synthesizing ribosomal proteins with antimicrobial activity based on the sequences of RPs previously identified as AMPs. This synthesis has been possible with several authors using the CAMP server (http://www.camp.bicnirrh.res.in/predict/, accessed on 21 June 2022) to predict the core sites for antimicrobial activity, thus identifying the antimicrobial core regions in different amino acid residues of these RPs. Moreover, homology analysis has revealed that the prokaryotes and eukaryotes in sponges, sea anemones, bivalves, starfish, fish, frogs, and mammals share a high homology, which has yielded speculation regarding the conservation of RPs with antibacterial characteristics throughout evolution. One of the first synthesized RPs with antimicrobial activity was described by Pütsep et al. [53], who cited that RP uL1 was composed of two amphipathic α-helices joined by a hinge. Unlike several RP uL1 proteins from other bacteria, the RP uL1 N terminus of *H. pylori* could form a perfect amphipathic helix; this first helix (residues 2–19) was followed by a second helix (residues 22–38). Thus, they synthesized and evaluated the antimicrobial activity of these peptides corresponding to the N-terminal part of RP uL1, of which, only the peptide called Hp (2–20), containing only the first α-helix of the two amphipathic α-helices, showed antimicrobial activity against *E. coli* and *Bacillus megaterium*. In the case of *E. coli*, some strains were enterohemorrhagic, indicating a food-borne pathogen that causes bloody diarrhea and, in some individuals, life-threatening hemolytic uremic syndrome (HUS) [68]. Furthermore, the research by Pütsep et al. [53] has been used in recent years as a basis for the synthesis of RPs with antimicrobial activity, due to the homology between RPs of different organisms.

One example of the homology between RPs and AMPs was published in 2010, where Khairulina et al. [69] reported that uS15 in eukaryotes and uS19 in prokaryotes exhibited homology. Subsequently, Qu et al. [1] published a study on the homology between the aforementioned RPs and the RP uS15 of amphioxi (*Branchiostoma japonicum*), which had antimicrobial activity against *Aeromonas hydrophila*, *E. coli*, *S. aureus*, and *B. subtilis*. These microorganisms present pathogenicity in fish as well as diarrhea, food poisoning, and skin and soft-tissue infections in individuals [70,71]. This evidence suggested that the antimicrobial activities of RPs could have an ancient origin and may have been highly conserved throughout evolution. To date, many RPs have been reported as being synthesized from the core regions of ribosomal proteins with antimicrobial activity; these are listed in Table 5.

**Table 5 ijms-23-09123-t005:** Different peptides with antimicrobial activity synthesized from ribosomal proteins.

Peptide	Peptide Sequence ^1^	Reference Organism	Against Microorganism	Physiochemical Properties		
				Hydrophobic	Iso-Electric Point (pH)	Net Charge at pH 7
RP uL1 (2–20) [53]	----	*H. pylori*	*E. coli*	----	----	----
RP uL1 HP-A3 (A3-NT) [72]	FKRLEKLFSKIWNWK-NH2	*H. pylori*	*C. albicans*, *Trichosporn beigelii*, and *Saccharomyces cerevisiae*	46.67%	11.3	11
RP uL1 (F1A) [73]	AKRLKKLFKKIWNWK-NH2	*H. pylori*	*E. coli*, *Pseudomonas aeruginosa*, *Proteus vulgaris*, *Salmonella typhimurium*, *S. aureus*, *L. monocytogenes*, *S. epidermidis*, *C. albicans*, *T. beigelii*, *Aspergillus awamori*, *Aspergillus flavus*, *Aspergillus fumigatus*, and *Aspergillus parasiticus*	46.67%	11.86	7
RP uL1 (F8A) [73]	FKRLKKLAKKIWNWK-NH2			46.67%	11.86	7
RP uL1 (F1AF8A) [73]	AKRLKKLAKKIWNWK-NH2			46.67%	11.86	7
RP uL1 (A2) [73]	AKRLKKLAKKIWKWK-NH2			46.67%	11.91	8
RP eL39 (PaDBS1R1) 2.1 kDa [74]	PKILNKILGKILRLAAAFK	*Pyrobaculum aerophilum*	*Klebsiella pneumoniae* and *S. aureus*	57.89%	11.79	5
RP S23 (BjRPS23 67–84) 15.8 kDa [75]	MGKPRGLRSARKLKDHRRQQRWHDKSFKKAHLGTAVKASPFGGASHAKGIVLEKIGVEAKQPNSAIRKCVRVQLIKNGKKITAFVPNDGCLNYIEENDEVLVSGFGRKGRAVGDIPGVRFKVVKVANVSLLALFKEKKERPRS	*B. japonicum*	*E. coli*, *A. hydrophila*, *S. aureus*, and *M. luteus*	37.76%	11.22	21.3
RP S23 (BjRPS23 17–38) 2.6 kDa [75]	RRQQRWHDKSFKKAHLGTAVKA		*S. aureus*	31.82%	11.93	6.2
RP S23 (BjRPS23 67–84) 2.1 kDa [75]	RKCVRVQLIKNGKKITAF		*E. coli*, *A. hydrophila*, *S. aureus*, and *M. luteus*	44.44%	11.57	5.9
RP uS15 16.9 kDa [1]	MADEQAALKKKRTFRKYTYRGVDLDQLLDMSSEQLMEMMKARPRRRFSRGLKRKHLALIKKLRKAKKECPALEKPEVVKTHLRNTVIVPEMIGSIVAVYNGKTFNQVEVKPEMIGHYLGEFSITYKPVKHGRPGIGATHSSRFIPLK	*B. japonicum*	*A. hydrophila*, *E. coli*, *S. aureus*, and *B. subtilis*	39.46%	10.82	18.4
RP uS15 (45–67) [1]	RRFSRGLKRKHLALIKKLRKAKK			34.78%	12.73	12.1
RP eL30 12.7 kDa [2]	----	*B. japonicum*	*A. hydrophila* and *S. aureus*	----	----	----
RP eL30 (2–27) [2]	KQKRKTMESINSRLQLVMKSGKYVLG			34.62%	11.2	6
RP eL30 (23–46) [2]	KYVLGLKETLKVLRQGKAKLIIIA			54.17%	10.88	5
RP bS1 (V10I) 1.1 kDa [76]	VTDFGVFVEI	*Thermus thermophilus*	*T. thermophilus*	60%	0.66	−2
RP bS1 (R23I) 2.6 kDa [76]	RKKRRQRRRGGSar#(A)GVTDFGVFVEI			30.77%	12.24	7
RP bS1 (R23T) 2.5 kDa [76]	RKKRRQRRRGGSar#(A)GVVEGTVVEVT			26.92%	12.24	7
RP bS1 [77]	----	*T. thermophilus*	*P. aeruginosa*	----	----	----
RP bS1 (R23R) 2.8 kDa [78]	RKKRRQRRRGGGGLHITDMAWKR	*P. aeruginosa*	*P. aeruginosa*	21.74%	12.51	9.1
RP bS1 (R23L) 2.6 kDa [78]	RKKRRQRRRGGGGITDFGIFIGL			26.09%	12.41	7
RP bS1 (R23F) [79]	RKKRRQRRRGGSarGVVVHI-Asi-GGKF-NH2	*S. aureus*	*S. aureus*, *P. aeruginosa*, *E. coli*, and *Bacillus cereus*	29.63%	12.89	10.1
RP bS1 (R2DI) [79]	RKKRRQRRRGGSarGLTQFGAFIDI-NH2			28%	12.51	8
RP bS1 (R23EI) [79]	RKKRRQRRRGGSarGVQGLVHISEI-NH2			24%	12.51	8.1

^1^ Residues that confer antimicrobial activity, as well as those that are presumed to have been highly conserved during evolution. # Alanine was used instead of sarcosine in the synthesized peptides.

## 4. Applications of AMPs

AMPs have attracted interest due to being low-molecular-weight proteins that have broad-spectrum antimicrobial and immune-modulatory activities against infectious bacteria (Gram-positive and Gram-negative), viruses, parasites, fungi, and tumor cells. In addition, these AMPs are a group of natural proteins present in animals, plants, insects, and bacteria. Consequently, they are “natural antibiotics”. Furthermore, because several methods have been developed to design new synthetic AMPs by modifying the sequences of innate antimicrobial peptides in various organisms, large-scale production at minimal cost is possible. The urgent need for developing alternative agents to control microbial diseases, especially those that are, or have become, antibiotic-resistant, is increasing, not only in the medical industry, but also in the food, animal husbandry, agricultural, and aquacultural industries [3,52].

Antibiotic-resistant bacterial strains are continuing to increase worldwide. The World Health Organization (WHO) has warned that in the future, infections may not respond to antibiotics. In recent years, epidemics and pandemics have revealed that public health is under global threat, and it will be necessary to create novel, effective antimicrobial agents [52]. The COVID-19 pandemic, declared in 2020, highlighted the importance and urgency of the search for new antivirus peptides [80]. Some AMPs have been approved by the United States Food and Drug Administration (FDA) or the European Union (EU) European Medicines Agency (EMA), because mandatory certification by these entities is required for the entry of these AMPs into the market and for their clinical use. Table 6 lists the AMPs that have been approved by the FDA, as well as their specific application and route of administration. Lastly, the combination of AMPs with antibiotics has been proposed as an effective strategy for eliminating multidrug-resistant bacterial strains and decreasing antibiotic doses in monotherapy [52].

Chemical preservatives have the potential to harm the human body; therefore, AMPs have been used as natural preservatives in food (e.g., cheese, yogurts, meat, wine, juice, etc.) because they are resistant to acids, alkalis, and proteases, and easily hydrolyze under high temperatures. The AMPs commonly used in food preservation are nisin and polylysine. Nisin is a bacteriocin produced by *Lactobacillus lactis*, LAB used as food preservatives, which are generally recognized as safe (GRAS), and are approved by the FDA. Although nisin can inhibit Gram-positive food-borne pathogenic and spoiling bacteria, it is ineffective on yeast and Gram-negative bacteria. Another bacteriocin is pedocin PA-1, produced by diplococcus, also used as a food preservative and as a growth inhibitor of *L. monocytogenes*, which can cause meat degradation [52,80,83]. Finally, an innovative packing method has been used in which AMPs have been added to the composition of packing materials, which could have potential applications in the food industry [80].

The emergence of antibiotic-resistant bacteria in animal products has become a major threat to public health and food safety. For this reason, a new antibacterial strategy should be used in the breeding and aquaculture of poultry, swine, and ruminants, because it could improve production performance and immunity and promote the intestinal health of animals intended for human consumption. AMPs have been shown to act as antiviral agents against viruses that infect animals, such as porcine epidemic diarrhea virus (PEDV), porcine transmissible gastroenteritis virus (TGEV), respiratory syndrome coronavirus (SARS-CoV), infectious bronchitis virus (IBV), and influenza A. NK-lysine peptides (NKLPs) have been shown to have an inhibitory effect on nodavirus, infectious pancreatic necrosis, and spring viremia carp virus, which are devastating to fish farming [52,80].

Another important aspect is the excessive and long-term use of chemical pesticides to prevent the damage to agriculture caused by plant insects and pathogens, which can also result in environmental pollution and damage to human health. Some peptides have been synthesized to inhibit the in vitro growth of phytopathogenic bacteria (*Pectobacterium carotovorum* and *Pectobacterium chrysanthemison*): iseganan, pexiganan, and the hybrid peptide cecropinmelitin CAMEL [52]. Finally, AMPs have been identified for potential use in the medical, food, animal husbandry, agricultural, and aquacultural industries; however, we have only listed a few to highlight their broad potential.

Regarding RPs as AMPs, the literature refers to the probable applications of these peptides. However, they are still under investigation for clinical application; clinical studies are required prior to their application. One possible application is that of RP eL39 (PaDBS1R1), a synthesized peptide with potential application as a therapeutic antimicrobial agent [74,76]. RP uL1 in *H. pylori* has also demonstrated strong anticancer and antibacterial activities, as well as antifungal and anti-parasitic activities [84]. Another RP is the RP uL1 produced from *B. tequilensis*, which showed anticancer properties, highlighting its advantage for use in selective toxicity against cancerous cells [61]. Finally, another three synthesized AMPs (RP bS1 (R23F); RP bS1 (R2DI); and RP bS1 (R23EI)) could be used as antimicrobial peptides against Gram-positive and Gram-negative bacteria resistant to traditional antibiotics. Such peptides could be effective against a wide range of bacteria and could prevent them from developing a resistance to treatment [79].

## 5. Possible RP Mechanism of Action as AMPs

The mechanism of RPs with antimicrobial activity is not yet fully understood. However, in the last decade, authors who have identified RPs as AMPs have proposed hypotheses and conducted experiments to try to understand their mechanism of action. Current hypotheses include that RPs could interfere with the ribosomal assembly of closely related bacteria [57].

RPs have been considered as new AMPs; therefore, their mechanism of action could be related to the mechanism of action of AMPs. The exact mechanism of AMPs has been highly debatable, but there is general acceptance that the main mode of action involves cell membrane perturbation and/or permeabilization. Regarding RPs, Irazazabal et al. [74] indicated that using a fragment of RP eL39 (PaDBS1R1) presented potent antimicrobial activity against bacteria and fungi when inserted into the lipid bilayer of a cell membrane. The authors suggested that it may be due to the amphipathic α-helix conformation of the RP, which promotes hydrophobic and electrostatic interaction with the membrane, where the positive face targets the anionic bacterial surface (phospholipids polar heads), and the hydrophobic residues interact with the hydrophobic core of the membrane (phospholipid tails). Therefore, alterations are induced in the cell membrane, such as membrane permeability, which later leads to cell lysis.

Some of the modes of action of AMPs include their interaction with, or insertion into, bacterial membranes, causing a scrambling of the normal distribution of lipids between leaflets of the bilayer, the formation of pores, and a loss of intracellular targets. Qu et al. [1] demonstrated that RP uS15 performs a combined action with the bacterial membrane through lipopolysaccharides (LPSs) and lipoteichoic acid (LTA), as well as membrane depolarization, and can also induce intracellular ROS in bacteria, which may eliminate potential pathogens via apoptosis/necrosis. In the same vein, Chen et al. [2] showed that RP eL30 can bind to LPSs, LTA, and peptidoglycan (PGN), and these bindings can result in the lethal depolarization of the membrane, as previously mentioned. Similarly, RPs can induce intracellular ROS production in bacteria, which has harmful effects on microbial components such as DNA, RNA, lipids, and proteins [85,86] (Figure 5).

## 6. Future Developments and Conclusions

Some AMPs have been approved by the FDA and recognized as GRAS, but few AMPs have been adopted for industrial applications. These antimicrobial peptides are from natural sources, such as bacteria, marine, plants, and insects. They have some disadvantages, such as causing damage to the cell membrane and hemolytic side effects in cells, increased production costs and technical limitations in production, limited stability depending on the pH, and ease of hydrolyzation by proteases [85]. Therefore, interest in synthesizing AMPs has increased, and these concerns have largely been eradicated by obtaining synthesized peptides with the following characteristics: high antimicrobial activity, low toxicity to mammalian membranes, high protease and environmental stability, and ease of access with low-cost production.

In summary, this review highlights the RPs that have antimicrobial properties, which is why they have been proposed as new AMPs from either natural or synthetic sources. Even though these RPs are highly selective to the membranes of bacterial cells, compared with mammalian cells, they are ideal molecules with great potential for application as novel antimicrobial agents. Currently, few, if any, clinical trials are being conducted on the use of RPs as AMPs. Although there have been studies examining their antimicrobial application and selective toxicity against cancerous cells, they have not yet been used in industrial applications.

However, the mechanism of RPs with antimicrobial activity is not yet fully understood. Consequently, additional research is required to further understand the mechanism of action of RPs as AMPs, especially regarding their differentiation in genera and species. Exploring their ancient origins and highly conserved status between organisms may also serve to shed light on their properties. In addition, further studies are needed to validate RPs as AMPs, and to determine their applicability as antimicrobial agents.

In conclusion, our objective was to provide an overview of ribosomal composition, structure, nomenclature, and function, and to highlight their importance as novel AMPs, particularly in terms of their proteins. RPs as novel moonlighting proteins and their participation in protein synthesis in the ribosome shows their potential to be involved in activities against antibiotic-resistant diseases and cancerous cells. This article has provided tables detailing the RPs identified in recent decades, as well as their possible applications as antimicrobial agents in the food, agriculture, animal husbandry, and pharmaceutical industries. Finally, we described the current hypotheses and data regarding their mechanisms of action against pathogenic bacteria.

## Figures and Tables

**Figure 1 ijms-23-09123-f001:**
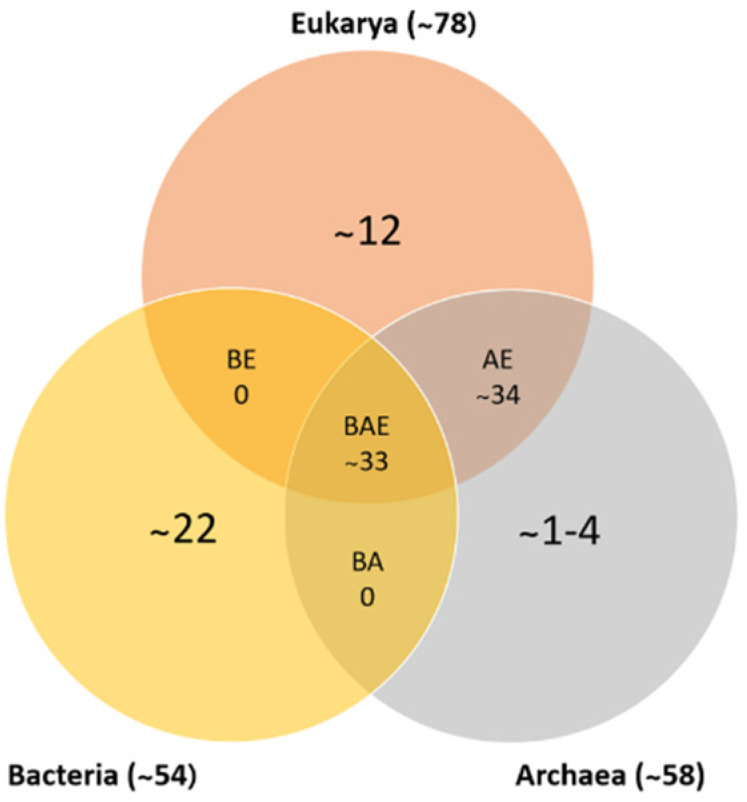
Numbers of ribosomal protein homologs shared between bacteria, archaea, and eukarya (BAE); bacteria and archaea (BA); archaea and eukarya (AE); bacteria and eukarya (BE); or unique (adapted from [10,11]).

**Figure 2 ijms-23-09123-f002:**
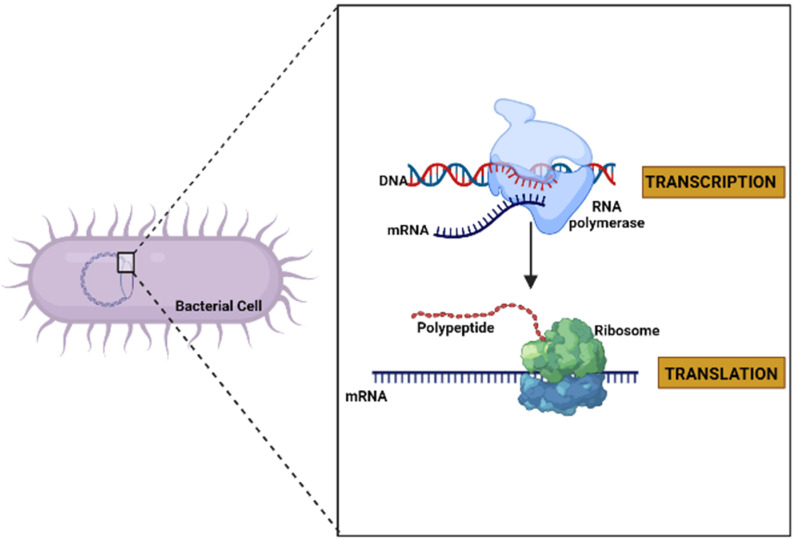
General diagram of the expression of a gene to protein through the processes of transcription and translation in a bacterial cell (created by Biorender.com) (accessed on 17 June 2022).

**Figure 3 ijms-23-09123-f003:**
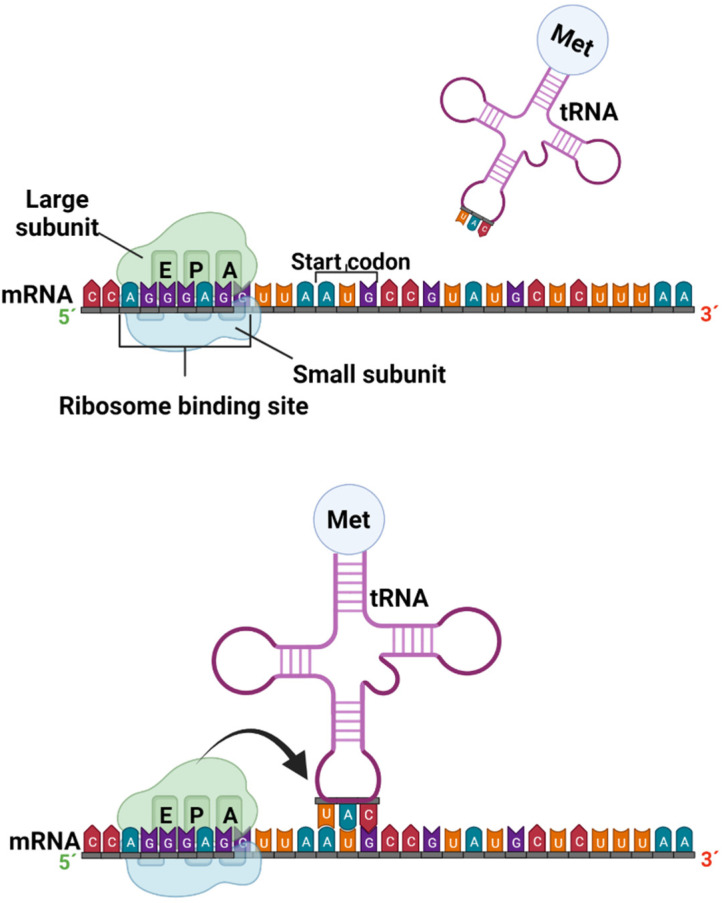
Formation of the preinitiation and initiation complex in the protein transcription process (created by Biorender.com) (accessed on 17 June 2022).

**Figure 4 ijms-23-09123-f004:**
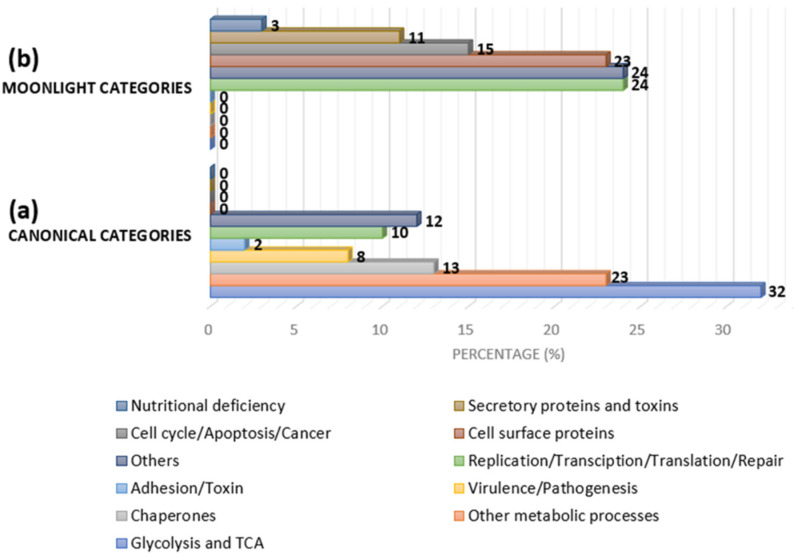
Approximate category distribution of MLPs based on canonical and secondary functions in bacteria: (**a**) 32% are glycolytic/tricarboxylic acid (TCA) cycle-related proteins, 23% are other metabolic pathway-related proteins, 13% are chaperones, 8% are virulence/pathogenesis-related proteins, 2% are adhesion- and toxin-related proteins, 10% are replication-, transcription-, translation-, and repair-related proteins, and the rest belong to other categories; and (**b**) 24% are replication-, transcription-, translation-, and repair-related proteins, 15% are cycle-, apoptosis-, and cancer-related proteins, 23% are cell-surface proteins, 11% are secretory toxin proteins, 3% are nutritional-deficiency-related proteins, and the rest are other categories (adapted from [44]).

**Figure 5 ijms-23-09123-f005:**
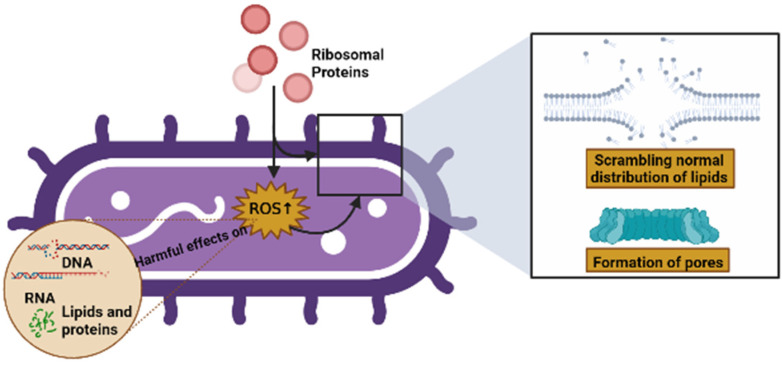
Possible causes of cell death caused by RPs (created by Biorender.com) (accessed on 17 June 2022).

**Table 1 ijms-23-09123-t001:** Ribosome composition in each domain of life [4,7,8].

	DOMAIN		
	Bacteria	Eukarya	Archaea
Ribosome	70S	80S	70S
Molecular mass	2.3 MDa	~4.5 MDa	~4.5 MDa
Large subunit (LSU)	50S	60S	50S
rRNA	23S, 5S	5.8S, 25–28S, 5S	23S, 5S
Number of proteins	33	46	40
Small subunit (SSU)	30S	40S	30S
rRNA	16S	18S	16S
Number of proteins	21	32	28

**Table 3 ijms-23-09123-t003:** Databases of moonlighting proteins (MLPs). Adapted from [43,44,45]. (accessed on 21 June 2022).

Database	Website
MoonProt	http://www.moonlightingproteins.org
Multitasking Proteins DataBase	http://wallace.uab.es/multitaskII
MultitaskProtDB-II	http://moondb.hb.univ-amu.fr
PlantMP	https://www.plantmp.com

**Table 6 ijms-23-09123-t006:** AMPs approved for clinical use by the FDA [52,80,81,82].

AMP	Application	Route of Administration
Dalbavancin, oritavancin, and telavancin	Complicated skin infections	Intravenous infusion
Vancomycin	Against Gram-positive bacteria Treats diarrhea associated with *Clostridium difficile*, *pseudomembranous colitis*, and infection	Intravenous infusion
Bacitracin	Skin and eye infections	Intramuscular
Polymyxin E (colistins)	Gastrointestinal tract infections caused by *E. coli* and *Salmonella* spp.	Intramuscular or intravenous
Polymyxin B	Last-line treatment alternative for resistant Gram-negative bacterial infections	Intramuscular, intravenous, intrathecal ophthalmic
Tyrothricin	Treatment of infected skin and oropharyngeal mucous membranes Effective against Gram-positive bacteria	Topical application only
Gramicidin D (or just gramicidin)	Skin lesions, surface wounds, and eye infections	External use only
Gramicidin S	Against Gram-negative and Gram-positive bacteria and fungi Used to treat genital ulcers caused by sexually transmitted diseases	Topical application only
Daptomycin	Skin infections caused by Gram-positive bacteria	Intravenous injection

## Data Availability

Not applicable.

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
