# Peer review of "Ribosomes: The New Role of Ribosomal Proteins as Natural Antimicrobials"

_ijms, 2022, doi:10.3390/ijms23169123_

Round 1
Reviewer 1 Report
Antibiotic resistance to bacterial pathogens is one of the most dangerous issues in the modern public health system. In the present review, Hurtado-Rios et al. discussed the usage of ribosomal proteins as a new intriguing alternative to the classical (small-molecule) antibiotics. It was shown that many ribosomal proteins may have an alternative, moonlighting or extra ribosomal function. One of them might be antimicrobial, antiviral and/or antiproliferative activities. In this review, the authors listed many examples of ribosomal proteins with the aforementioned activities as well as discussed the protentional mechanisms of their activity. Since the moonlighting functions of ribosomal proteins may have a different applications in medicine, pharmacology and nutritional science, this review will be interesting for the broad auditorium. Although, I would recommend it for publication after minor revision.
The authors did a great job by providing all important for understanding background information, part two of this review (“Ribosome”) seems to be too long. I would suggest shortening this part significantly. For instance, the discussion of the Svedberg coefficient (lines 59-74) falls beyond the topic of this review. Figures 3-5 also describe concepts which are not directly related to the review's primary topic.
In many places of the review, the nomenclature of ribosomal proteins is confusing. I would suggest sticking to wildly accepted nomenclature (citation 15 of this review). For instance, the authors name in table 2 of this review, protein “S2b” etc. But according to the nomenclature from citation 15, it should be “uS2” etc. Moreover, the nomenclature used by the authors is even more confusing in the text. For example, line 333 of the manuscript, says “40S RP S30”. The label “40S” is redundant because there is no “60S RP S30” protein. The same is valid for paragraphs 3-5 and tables 4 and 5.
I would also suggest adding more citations to the text of paragraphs 3-5. Large parts of the text there missed critical citations. For instance, lines 302-318 have only two citations. At the same time, it misses the important citation after the second sentence of the paragraph etc.
In general, the manuscript is clearly written and all important illustrations are present. It is important to mention that the authors summarized a lot of information in the table format. Although this manuscript in my opinion requires a revision, it should not preclude publication in the International Journal of Molecular Sciences.
Author Response
We appreciate all the constructive comments on the first version of the manuscript and we believe that the revised version fulfils all points mentioned.
Comment 1: Part two of this review (“Ribosome”) seems to be too long. I would suggest shortening this part significantly. For instance, the discussion of the Svedberg coefficient (lines 59-74) falls beyond the topic of this review. Figures 3-5 also describe concepts which are not directly related to the review's primary topic.
Response: According with the observations the section regarding the ribosome structure was reduced, in particular the discussion of the Svedberg coefficient was deleted. Also, the Figure that described the ribosome structure was deleted. The modified section corresponds to lines 45 – 193.
Comment 2: In many places of the review, the nomenclature of ribosomal proteins is confusing. I would suggest sticking to wildly accepted nomenclature (citation 15 of this review). For instance, the authors name in table 2 of this review, protein “S2b” etc. But according to the nomenclature from citation 15, it should be “uS2” etc. Moreover, the nomenclature used by the authors is even more confusing in the text. For example, line 333 of the manuscript, says “40S RP S30”. Thelabel “40S” is redundant because there is no “60S RP S30” protein. The same is valid for paragraphs 3-5 and tables 4 and 5.
Response: The nomenclature of ribosomal proteins was rewritten through the manuscript, including tables 2, 4 and 5, based on the assignment of homologous ribosomal proteins with the same name, irrespective of species, followed by the prefixes “b”, “a”, and “e”, as well as the prefix “u” that indicates if the protein is present in the ribosomes of the three domains of life. As the following references:
Ban, N.; Beckmann, R.; Cate, J. H.; Dinman, J. D.; Dragon, F.; Ellis, S. R.; Lafontaine, D. L. J.; Lindahl, L.; Liljas, A.; Lipton, J.M.; et al. A new system for naming ribosomal proteins. Curr. Opin. Struct. Biol. 2014, 24, 165-169; DOI: 10.1016/j.sbi.2014.01.002
Mitroshin, I. V.; Garber, M. B.; Gabdulkhakov, A. G. Investigation of structure of the ribosomal L12/P stalk. Biochem. (Mosc). 2016, 81, 1589-1601; DOI: 10.1134/S0006297916130022
Comment 3: I would also suggest adding more citations to the text of paragraphs 3-5. Large parts of the text there missed critical citations. For instance, lines 302-318 have only two citations. At the same time, it misses the important citation after the second sentence of the paragraph etc.
Response: We added the following citations [1, 2, 23-25, 30-41] within in the paragraph in lines 280 – 348.
Reviewer 2 Report
The review entitled "Ribosomes: the new role of ribosomal proteins as natural anti- microbials” written by Hurtado-Rios et al, focused on the potential function of moonlight proteins and their roles as an anti-microbial, fungal, and tumor as well. The authors initially more focused on ribosome structure(Line 65-7) in the S unit and in my opinion in this review it was not necessary. Later, they were focused on basic translational mechanisms and pre-initiation complex but still not focused on their main topics. So, I suggest authors improve their manuscript and more focused on moonlight protein function, regulation, and actions instead of ribosome structure and translation. Therefore, in my opinion, the manuscript is not ready for publication in its current form
Author Response
We appreciate all the constructive comments on the first version of the manuscript, and we believe that the revised version fulfils all points mentioned.
Comment 1: The authors initially more focused on ribosome structure (Line 65-7) in the S unit and in my opinion in this review it was not necessary.
Response: The section regarding on ribosome structure was revised and summarized, in lines 64 – 181 and 144 – 171. Also, the S unit was deleted.
Comment 2: Later, they were focused on basic translational mechanisms and preinitiation complex but still not focused on their main topics.
Response: According with the observations, the topic about the translational mechanisms and preinitiation complex was revised and rewritten in the manuscript, bearing in mind that is not the main topic of the article but leaving a general idea of the main process in which ribosomal proteins participate.
Comment 3: I suggest authors improve their manuscript and more focused on moonlight protein function, regulation, and actions instead of ribosome structure and translation.
Response: We added information about the protein function and regulation of moonlighting proteins, within in the paragraphs 4, 5 and 6 that comprehend the lines 215 – 257 and summarized the part of ribosome structure and translation.
Reviewer 3 Report
The authors introduced what are ribosomal proteins and summarized examples of ribosomal proteins and the peptide they synthesized that have antimicrobial activity. This review could provide an overview of the ribosomal protein and peptides potential used as antimicrobial drugs applied in pharmacy, agriculture, and food industry. There is one point that could trenghthen the manuscript.
In the fourth part, the authors could summarize the ongoing clinical trial of RPs and the successful cases of RPs in other industries.
Author Response
We appreciate all the constructive comments on the first version of the manuscript, and we believe that the revised version fulfils all points mentioned.
Comment 1: There is one point that could trenghthen the manuscript. In the fourth part, the authors could summarize the ongoing clinical trial of RPs and the successful cases of RPs in other industries.
Response: According with the observation, we included clinical trials of RPs and their successful cases in the industry in lines 419 - 498. In addition, Table 6 summarizes the ongoing clinical trials of RPs.
Round 2
Reviewer 2 Report
The authors improved the manuscript sufficiently.
Author Response
The authors appreciate all constructive comments, and believe that the revised version fulfils all points mentioned.